# Occipital Nerve Stimulation for Pain Modulation in Drug-Resistant Chronic Cluster Headache

**DOI:** 10.3390/brainsci11020236

**Published:** 2021-02-13

**Authors:** Javier Díaz-de-Terán, Javier A. Membrilla, José Paz-Solís, Iñigo de Lorenzo, Javier Roa, Manuel Lara-Lara, Alfonso Gil-Martínez, Exuperio Díez-Tejedor

**Affiliations:** 1Neurology Department, University Hospital La Paz, 28046 Madrid, Spain; javierddt@gmail.com (J.D.-d.-T.); membrillaja@gmail.com (J.A.M.); pinigo2593@gmail.com (I.d.L.); javier.roa.escobar@gmail.com (J.R.); manuellaral@hotmail.com (M.L.-L.); exuperio.diez@salud.madrid.org (E.D.-T.); 2CranioSPain Research Group, Departamento de Fisioterapia, Centro Superior de Estudios Universitarios La Salle (UAM), La Salle Campus Madrid, 28023 Madrid, Spain; 3La Paz Institute for Health Research (IdiPAZ), 28046 Madrid, Spain; 4Neurosurgery Department, University Hospital La Paz, 28046 Madrid, Spain; jfpaz@telefonica.net; 5Unit of Physiotherapy, University Hospital La Paz, 28046 Madrid, Spain

**Keywords:** cluster headache, chronic, refractory, drug-resistant, neuromodulation, occipital nerve stimulation

## Abstract

Occipital nerve stimulation (ONS) is a surgical treatment proposed for drug-resistant chronic cluster headache (drCCH). Long-term series assessing its efficacy are scarce. We designed a retrospective observational study with consecutive sampling, evaluating the follow-up of 17 drCCH patients who underwent ONS. Our main endpoint was the reduction the rate of attacks per week. We also evaluated the pain intensity through the Visual Analogue Scale (VAS), patient overall perceived improvement and decrease in oral medication intake. After a median follow-up of 6.0 years (4.5–9.0), patients decreased from a median of 30 weekly attacks to 22.5 (5.6–37.5, *p* = 0.012), 7.5 at 1 year (*p* = 0.006) and 15.0 at the end of follow-up (*p* = 0.041). The VAS decreased from a median of 10.0 to 8.0 (*p* = 0.011) at three months, to 7.0 (*p* = 0.008) at twelve months and 7.0 (*p* = 0.003) at the end of the follow-up. A total of 23.5% had an overall perceived improvement of ≥70% at 3 months, 41.2% at 1 year and 27.8% at the end of follow-up. Reducing prophylactic oral medication was possible in 76.5% and it was stopped in 17.7%. Triptan use decreased in all the responder patients and 17.7% stopped its intake. A total of 41.2% presented mild adverse events. In conclusion, our long-term experience suggests that ONS could be an interesting option for drCCH-selected patients, as it is a beneficial and minimally invasive procedure with no serious adverse events.

## 1. Introduction

Cluster headache (CH), the most common trigeminal autonomic cephalalgia, is defined as the onset of severe but strictly unilateral pain, which can be orbital, supraorbital, temporal or any combination of these, lasting 15–180 min, with a frequency of 1–8 per day, accompanied by a sense of restlessness and autonomic features in the trigeminal area [1]. CH is classified into episodic and chronic forms based on whether cluster-bout periods are separated by pain-free intervals (out-of-bout periods) that last at least 3 months [2]. Typically, CH patients respond to the usual treatments; however, in some cases, they become drug-resistant chronic CH (drCCH) [3].

For those patients who are refractory to the standard treatments, neurostimulation is an encouraging opportunity. Sphenopalatine ganglion stimulation, occipital nerve stimulation (ONS) and deep brain stimulation of the hypothalamus have all been proposed as invasive neuromodulation treatments for cluster headache [4]. Within the options of neurostimulation, the ONS is a recommended treatment option according to different European guidelines and consensus [5,6].

The ONS is one of the safest and most attractive choices for patients suffering drCCH, given the risk of serious complications described after other advanced therapeutic interventions. For example, deep-brain stimulation (DBS) may cause the development of contralateral cluster headache, seizure due to electrode repositioning after trauma, deep electrode infection leading to sepsis, death and ventricular hemorrhage [7]. This therapy is effective at targeting peripheral structures implied in CH pathophysiology, reducing the frequency and intensity of attacks through the inhibition of nociceptive activity in c-fibers and a-delta fibers. PET studies have also proved the central mechanism of ONS to be effective at normalizing hypermetabolic brain areas implicated in the pain matrix [8].

Invasive ONS has been assessed for CH prevention in different small open-label studies [8,9,10,11]. According to the American Headache Society’s evidence-based guidelines, there are not enough studies of occipital nerve stimulation, but the existing data suggest a benefit [12]. Due to the lack of strong supporting evidence for use of the ONS in patients with CCH, we want to contribute our long-term experience in the Headache Unit of a third level hospital.

The primary objective of this study is to evaluate the result of ONS as a long-term therapy for patients with drCCH, with the reduction in weekly headache attacks at the end of follow-up as our primary endpoint.

## 2. Materials and Methods

### 2.1. Study Population

This was a retrospective, observational study with consecutive sampling carried out at the Neurology Department of a third-level hospital in Madrid, Spain. We conducted a review of accumulated patient data obtained during the course of standard medical practice. No prospective treatment assignments were made, and all assessments were made through routine standard of care. Patients consented to individually giving their clinical history and headache diary retrospective data collection prior to ONS surgery. The study was reviewed and approved by the local Ethics Committee for Clinical Research of a public reference hospital (PI-4996). All patient data were treated with confidentiality, in fulfillment of the Declaration of Helsinki [13].

All patients included were diagnosed as chronic CH by a headache specialist, according to the International Classification of Headache Disorders 2nd edition (ICHD-II) (code 3.1.2.) [14], and fulfilled the European Headache Federation diagnostic definition of refractory CCH [3]. All patients had brain magnetic resonance imaging performed, with results within normal limits. These patients were considered candidates for ONS, following the recommendations of the Spanish Society of Neurology [6]. Prior to ONS implantation, all patients were evaluated by a multidisciplinary team of headache specialists composed by neurologists, neurosurgeons, anesthesiologists and psychiatrists. Implant surgery took place from March 2008 to June 2020. A minimum follow-up of 2 years after definitive ONS implantation was necessary for inclusion. A flow-chart of patient selection is available in Figure 1.

### 2.2. Variables and Definitions

According to the usual clinical practice in our Headache Unit, patient evaluation was performed before and every 3 months after surgery. Demographic data, number of preventive oral treatments trials and other therapeutic techniques before ONS indication and nosological information, were collected. Headache history (frequency, duration and intensity), clinical features, analgesics and triptans intake were also measured. Attack frequency (days of pain per week, month and number of attacks per day) and intensity (measured through the visual analogue scale (VAS)) were compared at baseline (prior to ONS implantation) and in the follow-up period. Overall perceived improvement was stratified in three groups depending on patient global improvement: poor responder < 30%; responder 30–50%; 50–70% good responder, super-responder ≥ 70%. Follow-up was defined as the period from ONS implantation surgery to system removal or deactivation. If the ONS was still active at the time of this study, it was considered an ongoing follow-up. 

Our primary endpoint was the reduction in the number of attacks per week at the end of the follow-up. To study the evolution of the change in this variable after treatment, an analysis was also carried out three months and one year after the intervention.

### 2.3. Data Collection

Data were collected retrospectively by checking medical reports.

### 2.4. Surgical Procedures

This technique requires two phases: a trial phase (implant of the electrodes) and a definitive implantation of the generator and the leads, if they were previously removed after the trial phase [15,16,17].

1.—Trial phase: In prone position, with mild sedation, a vertical incision 1 cm above and below the occipital protuberance in the midline is practiced. Two eight-contact electrodes can then be implanted bilaterally through a Tuohy needle (with ultrasound or x-ray control) in an epiphascial fashion, covering the major and minor occipital nerves. The patient feels a paresthesia that radiates towards the vertex bilaterally. If the electrodes are to be left in the trial phase, the incision is lengthened to 4 cm and the leads are anchored in the fascia, leaving an external cable that will be tunneled subcutaneously along the cervical midline to a mid-dorsal level. Once the trial phase has passed (2–7 weeks), and if it is successful (pain improvement > 50%), the leads are removed, or the external cable is cut, if it has been tunneled, leaving the leads implanted;

2.—Definitive phase: This can be done under general anesthesia (which is more comfortable for the patient) or with sedation. The patient can then be placed in prone position if the electrodes have been left previously, the nuchal incision is opened and the cable that was previously cut is removed. Another extension cable is connected to the buttock or abdominal region, where it is connected to the rechargeable generator. If the leads and generator are going to be placed all the way, the Trentmant technique is used [18], in a supine position, with the head lateralized, a medial and retro-mastoid incision is done, and the electrodes that are tunneled to the sub-clavicular region (there is no extension cable) are placed.

The electrodes are eight-contact electrodes, with a separation between contacts of 4 mm, so that the stimulating electric field is wider. The implanted generators are rechargeable devices with an independent current control manufactured by Boston Scientific, CA. Stimulation parameters are programmed contacts, 3–4 cathodes and 1–2 anodes, pulse width of 250 us and frequency of 60 Hz. The stimulation is continuous, although the patient can change the intensity of the stimulation on demand, according to his/her needs. The two sides of the head can be independently adjusted by the patient.

### 2.5. Statistical Analysis

Data analysis was performed using the Statistics Package for Social Science (SPSS 23.00-IBM, NY, USA). Nominal variables were reported as percentages and compared using a two-tailed Chi-square or Fisher test, when applicable. A two-tailed Shapiro–Wilk test for fewer than 30 participants was applied to examine whether the continuous quantitative variables followed a Gaussian distribution. Continuous quantitative variables were reported as mean ± standard deviation (SD) if they followed a Gaussian distribution; otherwise, they were represented as median ± interquartile range (IR). The Wilcoxon signed rank test was used to compare the median of two independent groups. If means were used, the independent samples *t*-test was used. Statistical significance was set at *p* < 0.05 and Confidence Interval was established at 95%.

## 3. Results

### 3.1. Patients Characteristics

As is depicted in Figure 1, twenty-two patients with drCCH underwent ONS trial, with 4/22 (18.2%) not presenting significant improvement and being dismissed for definitive surgery. Clinical characteristics of these patients are shown in Table 1. The remaining 18/22 patients (81.8%) had a positive result in the trial phase and proceeded to the definitive phase. One of them did not have a minimum follow-up of 2 years (the definitive ONS surgery was performed 4 weeks before this study). Of the lasting 17 patients, 10 were men (58.8%). Mean age at the time of surgery was 41.2 ± 8.7 years. Patients with CH had a mean of 12.7 ± 12.6 years since diagnosis. All patients were drug-resistant and 4.5 ± 1.7 oral preventive treatments were tried prior to ONS surgery indication according to current guidelines [5,6]. All patients underwent treatment trial with maximum tolerated doses of verapamil, topiramate, lithium and at least one more preventive oral drug (baclofen, valproic acid), as well as oral steroids. Greater occipital nerve (GON) infiltration was performed in 9/17 (52.9%), four of them and 4/17 (23.5%) were treated with onabotulinum toxin A (OnabotA) as well. OnabotA without GON infiltration was tried in 2/17 (11.8%). One patient (5.9%) was treated with transcutaneous electrical nerve stimulation and radiofrequency thermocoagulation of the sphenopalatine ganglion prior to ONS surgery. No patient underwent sphenopalatine ganglion stimulation. Further data are displayed in Table 2.

The median of follow-up after ONS surgery was 6.0 years (4.5–9.0). In 8/17 (47.1%) patients, the ONS is still activated and follow-up is ongoing. The remaining 9/17 (52.9%) patients had their follow-up ended by ONS deactivation (2/17, 11.8%) or removal (7/18, 41.2%). See Table 3 for further data.

### 3.2. ONS Effect on Number of Weekly CH Attacks

ONS successfully decreased the number of weekly CH attacks (see Figure 2). Prior to surgery, the median number of attacks per week was 30.0 (18.8–60.0). Three months after this intervention, the median number of weekly attacks decreased to 22.5 (5.6–37.5, *p* = 0.012). The maximum of the median number of attacks reductions was registered one year after ONS surgery, accounting for 7.5 attacks per week (1.6–41.3, *p* = 0.006). At the end of follow-up, the weekly number of CH attacks had a median of 15.0 (0.4–33.8, *p* = 0.041). A total of 3/17 (17.7%) patients had a cluster headache become episodic at the end of the follow-up.

### 3.3. ONS Effect on Pain Severity

The evolution of VAS of CH attacks after ONS was favorable. From a median of 10.0 (9.5–10.0) at baseline, VAS decreased after surgery to 8.0 (6.0–10.0, *p* = 0.011) at three months, to 7.0 (5.0–10.0, *p* = 0.008) at twelve months and 7.0 (5.0–9.8, *p* = 0.003) at the end of follow-up.

### 3.4. Overall Perceived Improvement

Patients were asked to rate their improvement after ONS surgery as under 30%, from 30 to 50%, from 50 to 70% or greater than 70% at different follow-up times (see Figure 3). The rate of super-responders (overall perceived improvement greater than 70%) was 23.5% (4/17) at 3 months and of 41.2% (7/17) after one year. An interesting fact is that 5/17 (29.4%) of individuals still were super-responsive at the end of follow-up. A more modest response (improvement of 30% to 50%) was seen in 4/17 (23.5%) both at 3 months and at 1 year, and 6/17 (35.3%) at the end of follow-up. Finally, the lesser improvement (under 30%) occurred in 8/17 (47.1%) at 3 months, in 6/17 (35.3%) at 1 year and in 7/17 (41.2%) at the end of follow-up.

### 3.5. Preventive Medication Use

All patients were under preventive medication at baseline. After this intervention, the improvement of 13/17 (76.5%) made it possible to decrease their prophylactic medication, mainly topiramate, verapamil and lithium, and 3/17 (17.7%) were able to stop all preventive drugs at the end of follow-up. A decrease in the need for corticotherapy was also observed. 

### 3.6. Triptan Use

Concerning the use of symptomatic medication, a decrease in subcutaneous sumatriptan was observed in all responders and 3/17 (17.7%) patients stopped triptan use. Median weekly triptan use in the responder group at baseline did not differ significantly from the non-responder group (14.0 (11.9–18.2) vs. 14.0 (9.8–21.0), *p* = 1.000). At the end of follow-up, responders achieved a lesser median triptan use per week (7.0 (2.8–7.3)), which did not occur in the group of non-responders (14.0 (9.1–18.6), *p* = 0.002).

### 3.7. Adverse Events

During follow-up, 41.2% (7/17) patients presented adverse effects. There was no mortality. All patients noticed mild paresthesia in scalp areas innervated by the occipital nerve. Stimulation parameters were set according to the patient tolerability. In the postintervention period, the most frequent AE was mild superficial surgical wound infections (2/17, 11.8%). Regarding long-term complications, 11.8% (2/17) presented electrode migration, one patient (4.6%) presented infection in the electrode area, another (5.9%) reported local pain at the electrode and another (5.9%) reported hardware dysfunction after trauma. Reintervention was needed for this patient and another eight (9/17, 52.9%) in this latest group, because of battery depletion, all experienced clinical worsening in terms of battery depletion and an improvement after its replacement. 

### 3.8. Non-Responders

At the end of follow-up, 6/17 (35.3%) were non-responders to ONS (i.e., the same or a greater number of weekly attacks). There was no statistically significant difference between non-responders and responders in terms of median CH duration (5.0 (0.3–18.3) vs. 9.0 (4.9–24.2), *p* = 0.462), median number of preventive medication used prior to ONS (3.5 (2.7–5.3) vs. 4.0 (3.6–6.1), *p* = 0.404), median weekly attacks at baseline (33.8 (10.0–67.5) vs. 30.0 (18.4–67.5), *p* = 1.000) and median years of follow-up (5.0 (2.5–8.2) vs. 7.0 (5.2–9.4), *p* = 0.256).

## 4. Discussion

Regarding our primary endpoint, the weekly attacks’ frequency, we observed a median of 15 attacks per week at the end of follow-up. The decline in this parameter improved progressively at the 3rd and 6th month and the greatest reductions were registered one year after ONS surgery, of 7.5 attacks per week, which reached 15 at the end of follow-up. The fact that ONS effectivity appears to slightly decrease over time was also reported by Aibar Duran et al. [19], but, in their study, the decrease in effectiveness was seen at month 6, and, in our study, it was found later. Despite this fact, our results are hopeful considering that this means a 50% reduction in attacks at the end of follow-up. As for the VAS, which presented a median of 10 at baseline, the greatest reduction in the intensity of the attacks was reached in the first year (VAS 7) and remained stable at the end of the follow-up (VAS 7). Concerning the overall perceived improvement, we also describe very good results 1 year after ONS implantation, reaching more than 70% patients with a response to the chronic stimulation. A mild reduction in benefit during longer follow-up, as in weekly attack number, was observed. Triptans and preventive medications were reduced or stopped in all responsive patients, which involves the achievement of a difficult therapeutic objective that has a direct impact on the patients’ quality of life. Considering all these parameters, it can be deduced that ONS achieves a sustained reduction in both the weekly attacks’ frequency and intensity, with a peak in response after one year of monitoring.

Prior to ONS, nine patients reported a positive response to occipital nerve block and, according to our results, as well as previous studies [20], this was not necessarily a predictor of good response to continuous stimulation with ONS. Of the nine patients who received a GON block, those who benefited from this procedure did not correspond with those who improved most with the ONS. As can be seen in Table 1, none of the patients who did not benefit from the trial phase had previously undergone the anaesthetic block.

During the first years of ONS implementation, several studies with short follow-up periods were published, showing variable results. In the study by Fontaine et al., 10/13 participants did not show a response until a median follow-up of 14.6 months [11]. Similar results were published by Magis et al. (78.6% were responders, median follow-up 36.8 months) [8]. On the other hand, Burns et al. described only 35.7% of responders, with a median follow-up of 17.5 months [10]. A recent meta-analysis included eight open studies with a sample of 96 patients, showing a response rate of 34–71% and a reduction in weekly attacks of 29% in a follow-up of 1–3 years [21]. The variability of these results could, in part, be due to the short follow-up and different definitions of responder patient. 

Several studies with a longer follow-up have been published in recent years. In their open-label study with 35 drCCH, Leone et al. described 66.7% of responders (50% reduction in headache number per day) after a follow-up period of 6.1 years [9]. In the second largest cohort with a prolonged follow-up (39.17 months), counting 51 patients, 52.9% exhibited a response to ONS at final follow-up. In the last months of 2020, the series with the largest cohort of patients to date was published [22]. At the end of follow-up (43.8 months), they reported a reduction in the frequency of attacks of > 50% in 69% of patients. Our results are more in line with these longer series than previous ones.

Considering that these patients are refractory to all therapies and suffer from high levels of disability and impact [23], a threshold of a > 30% decrease in the frequency of attacks is usually considered sufficient to define response according to different studies [24]. In our cohort, at the end of follow-up, and taking the 18 of the 22 patients who were implanted with a definitive ONS into account, 61.1% (11/18) of the patients presented a perceived improvement of more than 30%. Furthermore, almost a third of patients (27.8%) were super-responders at the end of follow-up. It is important to highlight these findings because they were drug-resistant patients who had gone through all kinds of therapies without obtaining any benefit, and three of them became almost asymptomatic, with no current preventive treatments and hardly any recurrence of attacks to date, while two other cases transformed into episodic CH.

An important issue that has been commented on in other similar studies is that the improvement in the patients could be related to the described natural evolution from chronic to episodic forms, as well as the placebo effect. However, it is unlikely that we encountered these possibilities, given the chronicity of our patients’ disease (12.2 ± 11.6 years) as well as the rapid worsening in the face of technical problems such as battery depletion or malfunction.

Although the ONS is not an economic treatment [25], the direct and indirect cost generated by the usually young patients diagnosed with drCCH widely exceeds the cost of this therapy [26]. Adverse events in our series were much lower than those described in previous cohorts [9,15,17,19,27]. ONS seems to be a safe technique when implants are conducted in highly specialised centers for those CH patients who have failed several preventive treatments, and we think it should be offered as the first option to drCCH patients.

Among the strengths of this study are a prolonged follow-up in a third-level hospital and the experience of a multidisciplinary headache team with extensive experience in neurostimulation. The main limitations of the present study include a small population being analyzed, a retrospective, single-center design and a lack of a control group. Another important limitation is the slight variability in the duration of the final follow-up of patients. It should also be noted that, technically, the ONS has evolved considerably in recent years. The more recent implanted devices are different in terms of size, battery life, and electrodes than those implemented at the beginning of the study in 2008.

Our data provide additional supporting evidence for the use of neuromodulatory therapies for drCCH treatment, although more prospective randomized, double-blind, multi-center clinical trials, like the ICON study [28], are needed to assess the effectiveness of ONS in the prevention and treatment for drCCH.

## 5. Conclusions

In summary, our long-term experience provides additional information about the long-term effect of ONS in patients with drCCH. This intervention appears to have a positive effect in decreasing attack frequency and all reported adverse events have been mild, which could make it a valid option when advanced treatment of CCH is required. Further research is needed to identify patients who are responders to the ONS in order to offer these therapies to properly selected cases.

## Figures and Tables

**Figure 1 brainsci-11-00236-f001:**
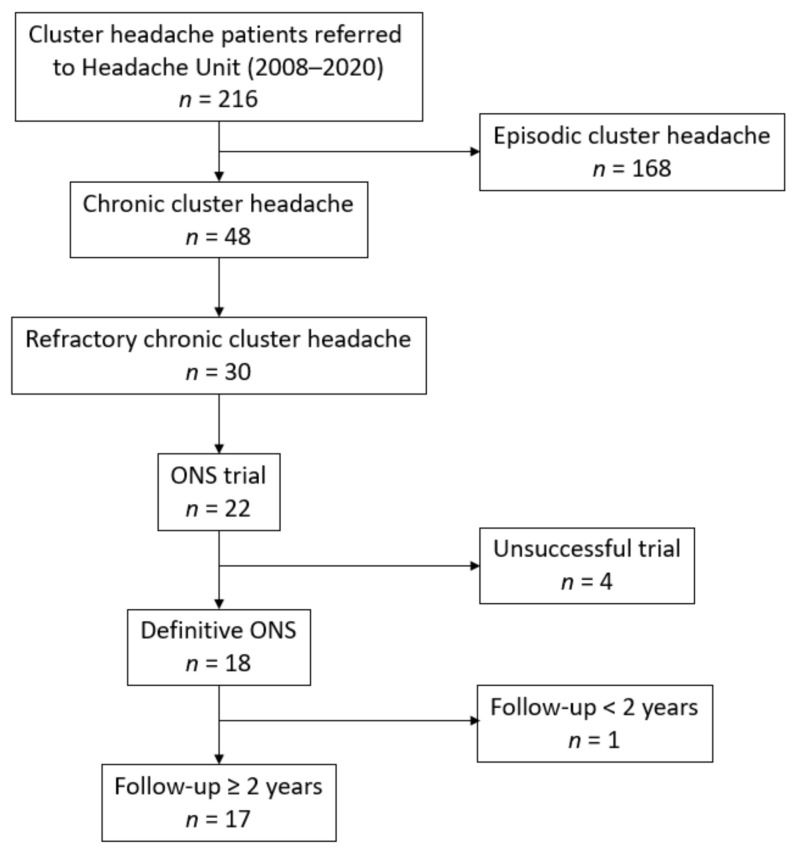
Flow-chart of patient selection (ONS: occipital nerve stimulation).

**Figure 2 brainsci-11-00236-f002:**
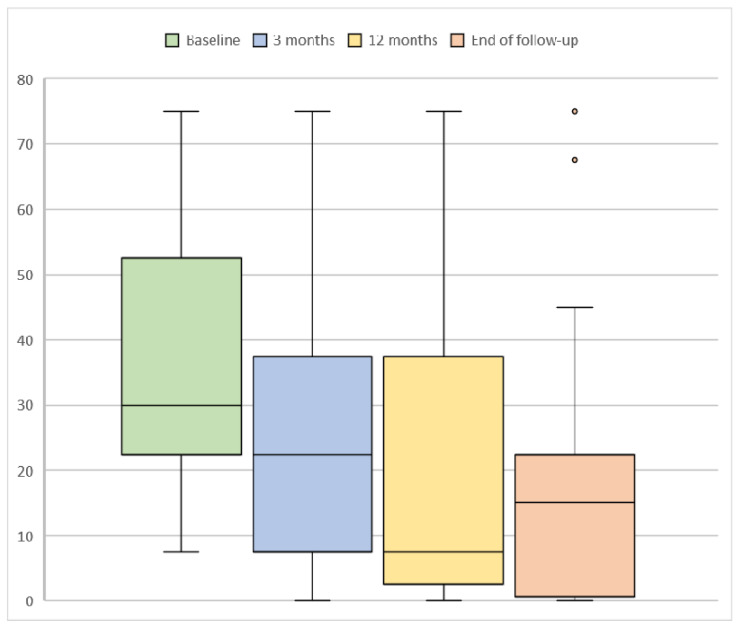
Evolution of weekly number of cluster headache attacks at baseline and after occipital nerve stimulation. Median and ranges are shown. *n* = 17 (no missing values between the groups). *p* values: 0.012 (baseline—3 months), 0.006 (baseline—12 months), 0.041 (baseline—end of follow-up).

**Figure 3 brainsci-11-00236-f003:**
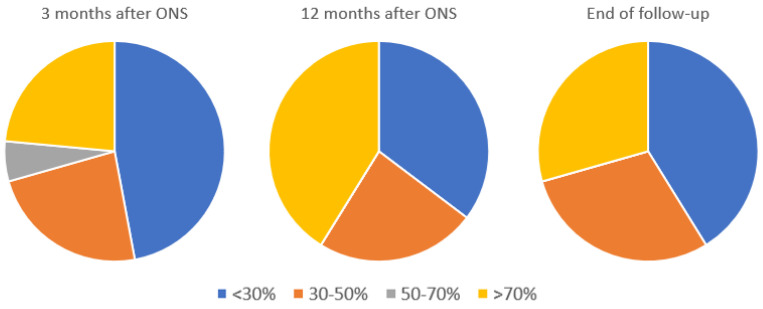
Overall improvement perceived by patients after occipital nerve stimulation (ONS) surgery. N = 17 (no missing values between the groups).

**Table 1 brainsci-11-00236-t001:** Epidemiological and nosological characteristics of cluster headache patients who did not improve in the occipital nerve stimulation trial.

Age	Gender	Duration of CH (Years)	Duration of Chronic CH (Years)	Preventive Oral Treatment Trials Prior to ONS	Therapeutic Techniques Prior to ONS	Prior to ONS Trial (Baseline)			Time of ONS Trial (Weeks)
						Days of Pain per Month	Attacks per Day	Attack VAS Score	
37	Female	3	1	3	None	20	1	8	3
34	Male	23	9	3	None	30	10	10	7
38	Female	14	11	3	OnabotA	30	8	10	3
36	Male	8	5	4	None	30	2	10	2

CH: cluster headache, ONS: occipital nerve stimulator, VAS: visual analogue scale, OnabotA: onabotulinumtoxin A.

**Table 2 brainsci-11-00236-t002:** Epidemiological and nosological characteristics of cluster headache patients treated with occipital nerve stimulator.

Patient No.	Age	Gender	Duration of CH (Years)	Duration of Chronic CH (Years)	Preventive Oral Treatment Trials Prior to ONS	Therapeutic Techniques Prior to ONS	Prior to ONS (Baseline)		
							Days of Pain per Month	Attacks per Day	Attack VAS Score
1	54	Female	10	7	4	OnabotA, GON TENS, SPG RFT, GON block	30	4	9
2	31	Female	4	2	3	GON block	30	1	10
3	36	Female	17	11	6	GON block	30	10 ¶	10
4	49	Female	31	22	6	GON block	30	1	10
5	45	Male	33	25	5	GON block	30	5	10
6	39	Female	23	16	3	None	30	9 ¶	8
7	61	Male	44	10	4	None	30	2	10
8	34	Female	2	1	5	None	30	4	9
9	37	Female	3	2	3	OnabotA	30	16 ¶	10
10	31	Male	4	3	4	None	10	1	10
11	41	Male	4	3	3	None	30	14 ¶	10
12	43	Male	4	2	8	OnabotA	30	3	10
13	42	Male	6	4	3	None	30	2	9
14	33	Male	4	1	3	GON block	30	7	10
15	31	Male	9	6	4	GON block, OnabotA	30	3	10
16	43	Male	7	5	8	Gon block, OnabotA	30	3	10
17	51	Male	11	6	5	GON block, OnabotA	30	5	10

¶ Total number of headache attacks per day in patient’s diary. Number of headache cluster attacks was 8, remaining headaches had migraine or tension-type headache characteristics. No.: number, CH: cluster headache, ONS: occipital nerve stimulator, VAS: visual analogue scale, OnabotA: onabotulinumtoxin A, GON: greater occipital nerve, TENS: transcutaneous electrical nerve stimulation, SPG: sphenopalatine ganglion, RFT: radiofrequency thermocoagulation.

**Table 3 brainsci-11-00236-t003:** Follow-up after occipital nerve stimulator.

Patient No.	Follow-Up *	Adverse Effects	3 Months after ONS				1 Year after ONS				At the End of Follow-Up			
			Days of Pain per Month	Attacks per Day	Attack VAS Score	Overall Perceived Improvement	Days of Pain per Month	Attacks per Day	Attack VAS Score	Overall Perceived Improvement	Days of Pain per Month	Attacks per Day	Attack VAS Score	Overall Perceived Improvement
1	4 years (deactivated)	None	3	1	3	>70%	3	1	3	>70%	9	30 µ	4	<30%
2	5 years (deactivated)	Electrode migration	1	1	10	>70%	1	1	10	>70%	30	1	10	<30%
3	5 years (removed)	None	30	10	10	<30%	30	10	10	<30%	30	10	10	<30%
4	12 years (ongoing)	Posttraumatic disconnection (reintervention needed)	0	0	NA	>70%	0	0	NA	>70%	Once every 3 months	Once every 3 months	3	>70%
5	9 years (ongoing)	Mild pain on surgery site	30	5	10	<30%	15	5	6	30–50%	Once every 3 months	Once every 3 months	6	>70%
6	8 years (removed)	None	30	9	7	<30%	30	9	7	<30%	30	9	7	<30%
7	10 years (ongoing)	None	30	2	7	30–50%	30	2	7	30–50%	30	2	7	30–50%
8	3 years (removed)	None	30	4	7	<30%	30	6	9	<30%	30	6	9	<30%
9	7 years (ongoing)	Electrode migration	30	5	6	30–50%	0	0	NA	>70%	0	0	NA	>70%
10	9 years (removed)	None	30	5	10	<30%	30	1	10	<30%	30	1	7	<30%
11	6 years (ongoing)	None	30	1	6	50–70%	30	10	8	30–50%	30	2	10	30–50%
12	5 years (ongoing)	Surgical site infection	30	1	6	30–50%	30	1	7	30-50%	30	2	8	30–50%
13	2 years (removed)	Surgical site infection	20	2	9	30–50%	30	2	10	<30%	30	2	10	<30%
14	12 years (ongoing)	None	15	1	5	>70%	5	3	5	>70%	3	1	5	>70% §
15	3 years (removed) #	None	30	3	10	<30%	10	3	5	>70%	Once every 6 months	1	5	>70%
16	5 years (removed)	Electrode infection	30	3	10	<30%	10	1	5	>70%	30	1	5	30–50%
17	7 years (ongoing)	None	30	5	10	<30%	30	5	10	<30%	30	3	3	30–50%

No.: number, ONS: occipital nerve stimulator, VAS: visual analogue scale, OnabotA: onabotulinumtoxin A, NA: not applicable. * Time from ONS placement to ONS deactivation or removal. Total number of headache attacks per day in patient’s diary. Number of headache cluster attacks was 8, remaining headaches had migraine or tension-type headache characteristics. µ Total number of headache attacks per day in patient’s diary. Number of headache cluster attacks was 8, remaining headaches had trigeminal neuralgia characteristics. § Cointervention: OnabotA started several years after ONS. # Removal performed by patient decision.

## Data Availability

Data available on request due to restrictions eg privacy or ethical. The data presented in this study are available on request from the corresponding author. The data are not publicly available due to privacy and ethical reasons.

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
