# Peer review of "Occipital Nerve Stimulation for Pain Modulation in Drug-Resistant Chronic Cluster Headache"

_brainsci, 2021, doi:10.3390/brainsci11020236_

Round 1
Reviewer 1 Report
The topic with ONS stimulation is very relevant and a long term follow up as the present authors describe is certainly of interest. However, there are several problems in the present version.
1) You include 22 patients and here 4 of them are only implanted in Phase 1, not permanently. So they should be regarded as drop outs/failures and taken out of the tables and the results, i.e. your sample is only 18 definitive ONS patients. Obviously the early results for those 4 should be reported and explained.
2) In the abstract You conclude that ONS is abeneficial treatment and there is no serious harm ( 41 % AE's??) and it should be offered as first option. This is far too strong conclusion and it is certainly not supported by your results and this open study. Please modify accordingly.
3) In the introduction You claim that "ONS is one of safest and most attractive treatments", BUT it is a strong argument without a reference and compared to what? SPG stimulation or DBS or?
4) Figure 1 Only 216 cluster patients in 12 years, it a very low number, why? please discuss very selected medically refractory or ?
5) In chapter 2.2 You write every 3 montsh after surgery... but later 3 mths after surgery and then 1 year and then at the follow up. please correct
6) 3 primary endpoints is not ok please choose one!
7)Please provide data on years with Chronic CH, not only disease duration as most patients develop from Episodic form
8) Take that patient out that had radiofrequency and SPG treatment, he is not eligible for this outcome! Here you write that it was before the ONS but claim later it was after ONS in the table?
9) 9 patients had experience with GON blocks, but later you say that it was succesful in 9 patients, not all GON blocks are succesfull and did they had repeatedly before and after ONS? Please clarify
10) What type and dose of medications had they tried please add this important information
11) In the table 1 and 2 you include all 22 patients but there is no mean or median line. only in the txt. please add and take the 4 patients out or mark them separately in Table 2, you cannot add weeks and years together here.
12) The marks below table 2 are not indicated in the table ?
13) In table 1 5/22 had more than 8 attacks per day, and up to 16/day is this Cluster headache or CPH? please clarify
14) Figure 2 should be revised after deselection of teh 4 patients and please only use the data from those that have provided all the time points and add an N=?, there are many missing values in the table.
15) chapter 3.2 it is Figure 2 You refer to not figure1.
16) please be precise and provide more data on the use of medication and their reduction (ie % and/or type).
17) a very important point, how many surgical revisions were there in the follow up? An AE percentage of 40,9% is high, so how safe is it?
18) there are no data on allodynia in the results or mentioned in method so take this part out.
19) How many became episodic after the treatment please add
20) The conclusions are very firm and strong and cannot be supported by the present dataset. Please moderate in the text and in the abstract
Reviewer 2 Report
The authors present a retrospective single center case series of drCCH patients treated with bilateral ONS. The follow up was between a few weeks up to 12 years, the benefit was very good in almost a fourth, and there were no serious adverse events.
In general the manuscript has a good structure but there are a few issues which have to been addressed.
1) methodological:
a) the variance of follow up is very large (from a few weeks to 12 years), therefore it is not very helpful to average the results of final follow up. The referee would suggest some other form of presentation: how many patients at 4 weeks, at 3months, a 1, 2, 5 and so on years, with a description of their averaged data. E.g. this would be in analogy with an ITT (intention to treat) vs. PP (per protocol) analysis.
b) 4 patients had a high frequency of attacks, CH criteria would not be fulfilled. In differential this might rather be paroxysmal hemicrania, please exclude.
c) you are presenting results in the methods section (page 4, lines 123-6)
d) please be clear in the AE section: why is wound infection acute and electrode migration and electrode infection chronic? Why do you count "expected" battery depletion into the AE in this case?
2) structure/content
a) figure 1: the total of 32544 is not appropriate for this flow chart. 164 and 48 do not add up to 216.
b) first paragraph on page 4 (again? second page 4), lines 280 ff. is totally hypothetical and somehow does not fit into the whole manuscript. please omit.
c) figure 3: why do you report data of only 17 patients?
Reviewer 3 Report
This paper is a report of a retrospective, single-center, observational study of 22 drug-resistant chronic cluster headache patients treated invasively with occipital nerve stimulation at a tertiary treatment center in Spain. Follow-up time period after start-up of treatment is greatly varying from 2 weeks to 10 years but those with follow-up time in weeks were those (with one exception) who only completed a trial phase and apparently did not have sufficient pain improvement to be judged to be treatment responders and therefore did not go on to the definitive treatment phase with cable extension to the abdominal region and generator implantation.
Though the study is small, retrospective and single center I do suggest that the scarcity of reports on this treatment suggests that the data reported is important, not least as this is a heavily afflicted patient group with few other treatment alternatives. In addition, other treatment alternatives are also invasive treatments and we need more information on all of them in order to compare and decide which one may suit individual patients best. Here, this paper reflects an important contribution for which I commend the authors. The authors report their results honestly and meticulously and the language and general presentation is clear and understandable.
The methodology, however, especially the retrospective nature of the study and the lack of a comparator group, limit the conclusions that may be drawn from the study. In addition, I have some comments on the interpretation of the data and the conclusion drawn. I hope the authors may find some suggestions on how to modify this in my comments.
I have the following comments:
Major points:
- The primary endpoint should be defined in the primary objectives section at the end of the introduction and should be very clear - it is now, in addition to in the discussion (line 231), given under variables and definitions (lines 97-98) where it is unclear which was the time point for the main assessment, with three different time points given. Was the primary end-point pre-decided? If the authors want to use all time-points and compare different times, then a time-series analyses should be used. This should then be considered and presented in the statistics section.
- My most central, overall comment is that the authors interpret their findings in a somewhat over-optimistic manner which leads to their conclusion of ONS as being beneficial and gives the impression (though not explicitely stated) that it is useful for the whole group of drug resistant cluster patients. This is based on several factors:i) perceived improvement does not include "no improvement" or, indeed, "worsening" which leads to a positive interpretation bias in the descriptive results (table 2); ii) trial phase results which are the basis for exclusion from the definitive phase are not reported; iii) the time for interpretation of the main outcome is not reported; iv) adverse effects including lead problems and infections are not sufficiently weighed in in the interpretation. To illustrate: A comparison of the descriptive data from all different follow-up time points (except after the trial phase) in table 2, with the baseline data in table 1 indeed shows that: after three months 6/18 patients have reported no subjective effect whatsoever (not just <30% effect), at one year this figure is 5/18 and at end of follow-up (with addition of the 4 patients that at the trial phase were perceived to be non-responders) 9/22 have no subjective effect. Regarding side effects 2/22 have electrode migration and 5/22 infections. These results suggest that there is a large group of non-responders which, given also adverse events risk, should not be offered this treatment. I suggest that describing non-responders and identifying them as such is important information which should carry greater weight in the manuscript.
- Why did non-responders still continue with their treatment as long as a mean of 5.1 years (range 2-9 years)?
- In the section on ethical permission etc (lines 66-68) it is not stated whether the patients were included by individual consent or their patient journals were addressed after such consent or not. Even if consent is not required in Spain for this type of retrospective study this should then at least be clearly stated.
- My suggestion is to modify the conclusion to more carefully state that ONS is positive in a subgroup of responders (see also major point 2 above) and that future prospective studies should aim to elucidate further how the responder (or non-responder) groups may be identified. The abstract should be modified accordingly.
Minor points:
- Figure 1. I suggest that rather than beginning each box with just a number this could preferably be placed after the box text in the form (n=48).
- The statistical analyses section should be expanded - what are "ratio variables" here? Paired tests? How was time-dependency of the main outcome variable addressed?
- Results (line 145) Open brackets in wrong place before "..were women".
- Table 2: add trial phase results
- Fig 2: ditto.
- Preventive med use (lines 213-216): I suggest this should be reported in more detail.
- Triptan use (lines 217-219): Report these numbers too.
- Discussion (line 231): Primary endpoint is not well defined (see above). Please also add time point for primary endpoint evaluation.
- Line 235: "mild trend" - please define and add to the methods section how this trend analyses was done, if there was no trend analysis, please rephrase.
- Lines 249-251: Even though it has been shown that response to nerve block is not a good predictor I would suggest you add response to this and response to the trial phase as this is used as your exclusion criteria for permanent stimulator implantation.
Round 2
Reviewer 2 Report
All raised comments have been addressed adequately.
The Quality of the manuscript has enlarged by much.
Reviewer 3 Report
Thank you to the authors for having made the suggested changes in most instances as well as having added additional explanations when unable to provide changes. In particular, I think that the present results and conclusions are more nuanced and that the broadened focus also to include a clearer focus on non-responders has improved the paper.
I commend the authors on their work. I have no further comments.